# *PINNsAgent*: Automated PDE Surrogation with Large Language Models

**Qingpo Wuwu** [1] [*]   **Chonghan Gao** [2] [*]   **Tianyu Chen** [2]   **Yihang Huang** [3]   **Yuekai Zhang** [1]   **Jianing Wang** [1]
**Jianxin Li** [2]   **Haoyi Zhou** [2]   **Shanghang Zhang** [1]

## Abstract

Solving partial differential equations (PDEs) using neural methods has been a long-standing scientific and engineering research pursuit. Physics-Informed Neural Networks (PINNs) have emerged as a promising alternative to traditional numerical methods for solving PDEs. However, the gap between domain-specific knowledge and deep learning expertise often limits the practical application of PINNs. Previous works typically involve manually conducting extensive PINNs experiments and summarizing heuristic rules for hyperparameter tuning. In this work, we introduce *PINNsAgent*, a novel surrogation framework that leverages large language models (LLMs) to bridge the gap between domain-specific knowledge and deep learning. *PINNsAgent* integrates Physics-Guided Knowledge Replay (PGKR) for efficient knowledge transfer from solved PDEs to similar problems, and Memory Tree Reasoning for exploring the search space of optimal PINNs architectures. We evaluate *PINNsAgent* on 14 benchmark PDEs, demonstrating its effectiveness in automating the surrogation process and significantly improving the accuracy of PINNs-based solutions. Project website: `https://qingpowuwu.github.io/PINNsAgent/`.

## 1. Introduction

Solving partial differential equations (PDEs) is a fundamental challenge with wide-ranging applications across various scientific and engineering domains, including fluid dynamics (Kutz, 2013), quantum mechanics (Teschl, 2014), and climate modeling (Stocker, 2011). Traditional numerical meth-

ods, such as finite difference (Strikwerda, 2004), finite element (Hughes, 2000), and finite volume methods (LeVeque, 2002), often incur significant computational costs, struggle to handle nonlinearities and complex geometries (Canuto et al., 2007; Berger & Oliger, 1984), motivating the development of data-driven alternatives. Physics-Informed Neural Networks (PINNs) have recently emerged as a promising deep learning-based approach for solving PDEs (Raissi et al., 2017; 2019). However, designing effective neural architectures for PINNs heavily relies on expert knowledge and often requires extensive trial-and-error to mitigate training pathologies (Wang et al., 2023b), prompting extensive research to explore optimal PINNs architectures and hyperparameters through numerous manual experiments. (Wang et al., 2022) conducted a comprehensive study to understand the relationship between PINNs architectures and their performance, while (Kaplarević-Mališić et al., 2023) manually explored various evolutionary strategies for PINNs architecture optimization. Similarly, (Wang & Zhong, 2024) investigated the impact of different architectural choices on PINNs performance through extensive experiments. These works often summarize their findings as thumb rules or guidelines for several PDE families to assist non-experts in manually tuning PINNs architectures and hyperparameters. However, this manual approach is time-consuming, labor-intensive, and may not generalize well to a wide range of PDEs, highlighting the need for an automated framework capable of hierarchically formulating and optimizing PINNs architectures for given PDEs.

Recent advancements in large language model (LLM)-based intelligent agents have showcased their capacity in the realm of scientific computing, automating tasks such as code generation (Nijkamp et al., 2023; Huang et al., 2023; Madaan et al., 2024; Wang et al., 2023c), hyper-parameter tuning (Zhang et al., 2023), and physical modeling (Alexiadis & Ghiassi, 2024; Ali-Dib & Menou, 2023). These agents leverage the vast knowledge encoded within LLMs to provide intelligent assistance and recommendations, opening up new possibilities for accelerating scientific research and development. However, their application in developing deep learning-based solvers for PDEs has yet to fully exploit the potential of leveraging existing database as foundational knowledge and experimental logs as iterative feedback. We

---

[*]Equal contribution  [1]State Key Laboratory of Multimedia Information Processing, School of Computer Science, Peking University  [2]School of Computer Science, Beihang University  [3]School of Artificial Intelligence, Beijing Normal University. Correspondence to: Shanghang Zhang <shanghang@pku.edu.cn>.

*Proceedings of the 42$^{nd}$ International Conference on Machine Learning*, Vancouver, Canada. PMLR 267, 2025. Copyright 2025 by the author(s).

aim to bridge this gap by developing an LLM-based intelligent agent (Brown et al., 2020) framework to autonomously optimize Physics-Informed Neural Networks (PINNs) architectures for solving Partial Differential Equations (PDEs) without relying on manual tuning or expert heuristics.

To this end, we introduce *PINNsAgent*, an innovative LLM-based surrogate framework that leverages LLMs as intelligent agents to develop and optimize PINNs autonomously. *PINNsAgent* comprises a multi-agent system, including a database that accumulates past experimental logs, a planner that generates candidate architectures and guides the exploration process, a programmer that translates designed architectures into executable code, and a code bank for storing and retrieving successful implementations. To efficiently utilize the knowledge stored in the database, we propose a new retrieval framework called Physics-Guided Knowledge Replay (PGKR) that encodes the essential characteristics of PDEs. Inspired by insights from existing literature (Wang et al., 2023b; 2022; Rathore et al., 2024; Saratchandran et al., 2024), we assign appropriate weights to different PDE features for better determining similarity. This weighted encoding enables efficient knowledge transfer from solved PDEs to similar problems by ranking their similarity scores. Additionally, to further explore and optimize the hyperparameter configurations provided by PGKR, we introduce the Memory Tree Reasoning Strategy (MTRS), which guides the planner in exploring the PINNs architecture space. This approach continuously improves PINNs architecture components via online learning with experiment feedback.

Evaluated on 14 diverse PDEs, *PINNsAgent* demonstrates superior performance compared to state-of-the-art methods, showcasing its ability to autonomously develop and optimize PINNs architectures. The critical contributions of our work can be summarized as follows:

- We propose *PINNsAgent*, a novel LLM-based surrogate framework that autonomously develops and explores optimal PINNs for given PDEs without relying on expert heuristics on deep learning. The framework consists of a multi-agent system, including a database, a planner, a programmer, and a code bank, which work together to generate and optimize PINNs architectures.

- We introduce Physics-Guided Knowledge Replay (PGKR), which encodes essential characteristics of PDEs and enables efficient knowledge transfer from solved PDEs to similar problems, significantly improving learning efficiency.

- We develop the Memory Tree Reasoning Strategy (MTRS) that guides the planner in navigating the PINNs architecture space, facilitating continuous improvement through iterative feedback.

## 2. Related Work

### 2.1. Learned PDE Solvers

Data-driven PDE solvers have garnered significant attention since (Raissi et al., 2017; 2019) first propose physics-informed neural networks (PINNs) to solve nonlinear PDEs by using automatic differentiation to embed PDE residuals into the loss function. However, vanilla PINNs exhibit various limitations in accuracy, efficiency, and generalizability, prompting extensive research towards developing improved differentiable neural network PDE solvers (Cuomo et al., 2022). For instance, (Yu et al., 2022; Wang et al., 2021) proposed loss functions that incorporate gradient enhancement of the PDE residual to improve model stability and accuracy. Another approach, as described (Jagtap et al., 2020b;a; 2022), introduced adaptive activation functions to reduce the inefficiency of trial and error in network training. Furthermore, to enhance computational efficiency and adapt to complex geometries, (Nabian et al., 2021; Shukla et al., 2021; Jagtap & Karniadakis, 2020) introduced importance sampling and domain decomposition. Despite these advancements, the selection and design of PINNs still pose barriers for non-experts. Our proposed *PINNsAgent* framework aims to provide an automated surrogate framework for proposing PINNs architectures to solve user-provided PDEs.

### 2.2. LLM-based Autonomous SciML Agents

Large language models have demonstrated powerful general knowledge and linguistic capabilities since the release of GPT-3.5(Ouyang et al., 2022), leading to various studies that extend their application towards specific tasks, known as large language model (LLM) agents. These agents have been applied to both general tasks and scientific disciplines (AI4Science & Quantum, 2023; Bran et al., 2024; Boiko et al., 2023). In the field of Scientific Machine Learning (SciML), various studies have highlighted the capability of LLMs to simulate physical phenomena, for instance, (Ali-Dib & Menou, 2023; Alexiadis & Ghiassi, 2024) demonstrate the use of LLMs in developing numerical PDE solvers, thereby accelerating scientific inquiries and discoveries. Additionally, (Kumar et al., 2023) combined LLMs with PDE solvers (PINNs and DeepONet), creating agents that assist in data preprocessing, model selection, and result interpretation. Similarly, (Lin et al., 2024) introduced a physics-informed LLM agent specifically designed for power converter modulation. However, these previous works have only preliminarily explored the potential of LLMs in solving PDE problems and conducted some case studies. Our work not only further validates the effectiveness of LLMs in these applications but also proposes a novel framework that seamlessly integrates LLMs with PINNs, contributing to the SciML community.

# PINNsAgent

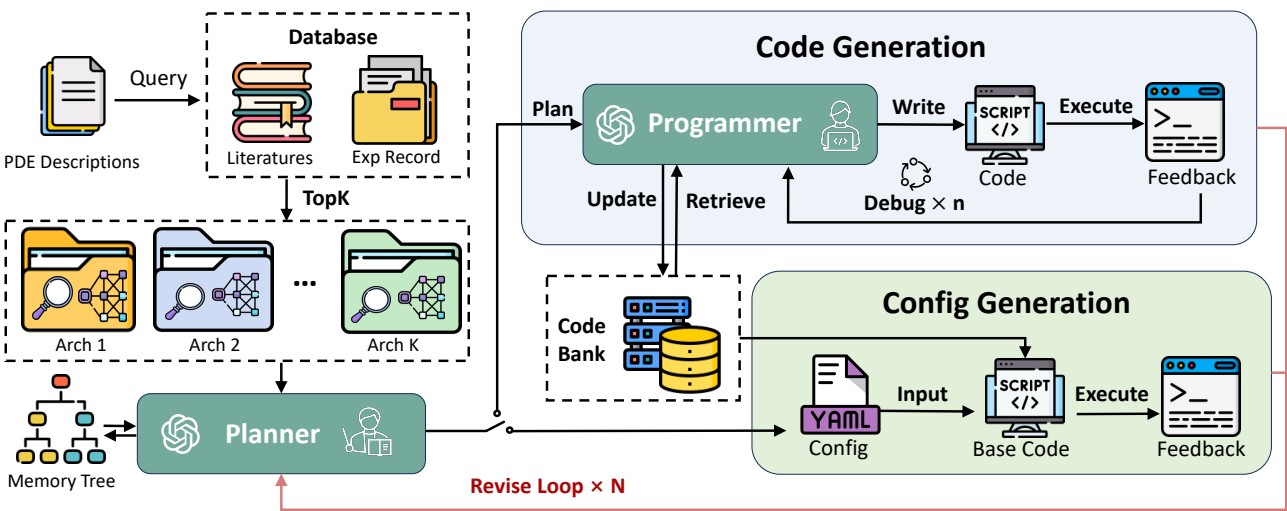

*Figure 1.* **The workflow of the *PINNsAgent*'s Framework.** The *PINNsAgent* operates in two modes: Code Generation and Config Generation. It leverages LLM agents to generate and refine executable code and YAML configuration files for optimizing hyperparameters in PINNs. The planner and programmer collaborate to devise experimental plans and generate training code, utilizing a central Code Bank and top-K cases from the Database.

## 2.3. LLMs enabled AutoML

Automated Machine Learning (AutoML) has revolutionized the field of machine learning by automating the selection of optimal models and their hyperparameters (He et al., 2021; Karmaker et al., 2021). Early approaches in AutoML focused on efficiently exploring hyperparameter spaces, a process known as Hyperparameter Optimization (HPO) (Feurer & Hutter, 2019). Representative methods include Random Search (Bergstra & Bengio, 2012), Grid Search (Liashchynskyi & Liashchynskyi, 2019), Bayesian Optimization (Wu et al., 2019), and Evolutionary Computation (Liu et al., 2023). With the advent of large language models (LLMs), HPO methodologies have significantly evolved. LLMs can automate and enhance HPO by generating predictive and insightful hyperparameter suggestions based on their extensive training data (Wang et al., 2023a; Tornede et al., 2024). Their reasoning capabilities allow them to propose initial hyperparameters by analyzing training tasks and datasets (Guo et al., 2024). Additionally, LLMs leverage cross-domain knowledge to improve model configurations and can generate parts or entire neural network architectures (Yu et al., 2023; Zheng et al., 2023). Our work differs from previous studies by introducing a novel multi-agent framework that automates the design of PINNs at two levels: database retrieval (PGKR), hyperparameter optimization (MTRS). This approach provides a balanced trade-off between efficiency and accuracy, offering significant improvements over

traditional methods.

# 3. Methods

In this session, we introduce *PINNsAgent*, a novel multi-agent framework for optimizing PINNs architectures.

Section 3.1 introduces the background of the proposed framework. Section 3.2 provides an overview of the critical components of *PINNsAgent*. In Section 3.3, we propose a novel retrieval framework called Physics-Guided Knowledge Replay (PGKR), which leverages the mathematical and physical properties of PDEs to identify promising hyperparameter configurations. Finally, Section 3.4 introduces the Memory Tree Reasoning Strategy within *PINNsAgent* for efficient exploration of the search space.

## 3.1. Preliminary

### 3.1.1. PROBLEM FORMULATION: SOLVING PDES WITH PINNS

Partial Differential Equations (PDEs) are foundational to modeling various physical phenomena across science and engineering disciplines, including fluid dynamics (Kutz, 2013), quantum mechanics (Teschl, 2014), and climate modeling (Stocker, 2011). A general form of a PDE is expressed as:

$$F(x, u, \nabla u, \nabla^2 u, \dots) = 0, \qquad (1)$$

where $u = u(x)$ denotes the unknown function, $x$ the spatial coordinates, and $\nabla u$, $\nabla^2 u$ the first and higher-order spatial derivatives of $u$.

Physics-Informed Neural Networks (PINNs) provide a mesh-free method to solve PDEs by utilizing the universal approximation capabilities of deep neural networks. PINNs enforce the compliance of the neural network solution $u(\theta, \mathcal{H})$ with the underlying physical laws represented by the PDEs. The overall formulation of a PINNs is given by:

$$\mathcal{L}(u(\theta, \mathcal{H})) = \mathcal{L}_{PDE}(u(\theta, \mathcal{H})) + \mathcal{L}_{BC}(u(\theta, \mathcal{H})), \quad (2)$$

where $u(\theta, \mathcal{H})$ is the neural network approximation of $u$, parameterized by weights $\theta$ and hyperparameters $\mathcal{H}$. The PDE-residual loss $\mathcal{L}_{PDE}$ is computed as:

$$\mathcal{L}_{PDE}(u(\theta, \mathcal{H})) = \frac{1}{N} \sum_{i=1}^{N} \left| F(x_i, u, \nabla u, \nabla^2 u, \dots) \right|^2. \tag{3}$$

where $N$ represents the number of collocation points used to evaluate the PDE residuals.

The boundary condition loss $\mathcal{L}_{BC}$ enforces agreement with the prescribed boundary (and/or initial) conditions, and is typically defined as:

$$\mathcal{L}_{BC}(u(\theta, \mathcal{H})) = \frac{1}{N_{BC}} \sum_{i=1}^{N_{BC}} \left| u_\theta(\mathbf{x}_i^{BC}) - u_{BC}(\mathbf{x}_i^{BC}) \right|^2, \tag{4}$$

where $N_{BC}$ is the number of boundary (or initial) sample points, $\mathbf{x}_i^{BC}$ denotes their coordinates, and $u_{BC}$ is the prescribed value at $\mathbf{x}_i^{BC}$.

### 3.1.2. HYPERPARAMETER OPTIMIZATION VIA LLM

Selecting optimal hyperparameters $\mathcal{H}$, such as learning rates, layer depths, neuron counts per layer, and activation functions, is crucial for the training efficacy and solution accuracy of PINNs. Traditional methods like grid or random search are often inefficient and computationally demanding. Large Language Models (LLMs) offer a novel iterative approach to generate hyperparameter settings. Formally, the LLM-based HPO method encompass an iterative loop:

$$p_{\text{LLM}}(\mathcal{H}^t) = \sum_{f^{t-1}} p_{\text{pr}}(f^{t-1}|\mathcal{H}^{t-1}) p_{\text{pl}}(\mathcal{H}^t|\tau, \mathcal{H}^{t-1}, f^{t-1}), \tag{5}$$

where $\mathcal{H}^t$ denotes the hyperparameters at iteration $t$, $\tau$ is the PDE formulation, and $f^{t-1}$ is feedback from the prior iteration. We introduce two agents: a planner $p_{\text{pl}}$ and a programmer $p_{\text{pr}}$. The planner generates new hyperparameter settings based on the problem formulation $\tau$, previous settings $\mathcal{H}^{t-1}$, and feedback $f^{t-1}$. The programmer executes training scripts for PINNs using the settings $\mathcal{H}^{t-1}$ and produces $f^{t-1}$. The initial setting $\mathcal{H}^0$ is defined as $p(\mathcal{H}^0) = p_{\text{R}}(\tau, B)$, where $p_{\text{R}}$ is a retriever querying a pre-established database $B$ with the PDE formulation $\tau$ to establish a starting point $\mathcal{H}^0$.

### 3.2. PINNsAgent

*PINNsAgent* is an LLM-based multi-agent framework that integrates three key components: the Database, the Planner, and the Code Bank. These components work collaboratively to develop optimal PINNs architectures for target PDEs.

As illustrated in Figure 1, the PINNsAgent operates in two distinct modes: Config Generation and Code Generation. The Config Generation mode, which is the primary focus of this study, is designed to handle scenarios where the target PDE already exists in the Code Bank. In this mode, the LLM-based agent, termed the planner, is tasked with generating YAML configuration files. These files delineate the settings within a predefined search space, optimizing the hyperparameters in accordance with the specific requirements and constraints of the target PDEs. The Code Generation mode is designed to address scenarios where the user specifies a PDE that is not present in the Code Bank, enabling PINNsAgent to handle user-specified PDEs that are new to the Code Bank.

### 3.2.1. STEP 1: DATABASE RETRIEVAL

The Database acts as a central repository for archiving both the literature related to PINNs and the successful hyperparameter configurations derived from prior experiments. To capitalize on this accumulated experience efficiently, we introduce a novel retrieval strategy named Physics-Guided Knowledge Replay (PGKR). Upon querying the Database with a detailed description of the PDE, select the top $K$ hyperparameter settings that best align with the requirements of the target PDE. These selected settings are then forwarded to the planner, which synthesizes this information to devise a comprehensive experiment plan. The details of PGKR are elaborated in Section 3.3.

### 3.2.2. STEP 2: EXPERIMENT PLAN GENERATION

In this step, the planner plays a pivotal role in generating candidate architectures and guiding the exploration of the hyperparameter search space for PINNs. Utilizing the top $K$ initial configurations sourced from the database, the planner functions as a policy model tasked with the strategic explo-

ration of the PINNs architecture search space. To navigate this search space efficiently, we have developed the Memory Tree Reasoning Strategy (MTRS), a novel method designed to optimize the selection process of hyperparameters by evaluating their exploration scores. The details of MTRS are thoroughly elaborated in Section 3.4. After selecting $\mathcal{H}^i$ using MTRS, the planner proceeds to develop a comprehensive experimental plan. This plan serves as a blueprint for the programmer to implement the PINNs according to the specified configurations. In the case of Config Generation mode where the programmer is not involved, the planner is required to generate the configuration files in YAML format, based on the feedback $l^{t-1}$ of the former iteration.

### 3.2.3. STEP 3: CODE EXECUTION

To facilitate the deployment of PINNs models, we construct a Code Bank to store the reusable code snippets, providing the programmer with successful examples and API instances. In the Code Generation mode, The programmer retrieves pre-defined templates, libraries, and best practices from the Code Bank and translates the candidate architectures generated by the planner into executable code following the experiment plan. If errors are reported, the terminal feedback is then replayed to the programmer to identify and resolve bugs. In the Config Generation mode, we extract the base code that can directly run on the generated configuration file to get the final results.

### 3.2.4. STEP 4: REVISION

Upon completing the execution step, the evaluation results $f^t$, including detailed training logs, performance metrics, and visualization results, are utilized to update the Database. This process ensures that each iteration enriches the repository with new insights and empirical evidence, contributing to a more comprehensive knowledge base. Using the feedback $f^t$, the Planner revises the experimental plan to enhance the model's performance in subsequent iterations. This feedback-driven revision process involves systematically adjusting the hyperparameter settings and potentially exploring new architectural modifications. The revision process is meticulously executed through a loop that iterates $N$ times. In each iteration, the Planner assesses the current performance, identifies areas for improvement, and makes informed adjustments to the hyperparameters.

### 3.3. Database Exploitation: Physics-Guided Knowledge Replay (PGKR)

Experience Replay is a paradigm for reusing past experimental data and expert knowledge to address new challenges (Rolnick et al., 2019). However, there is a lack of sufficient databases and literature supporting the retrieval of optimal architectures for PDEs in database components when de-

signing PINNs.

To address this issue, we conducted 3000 parameter fine-tuning experiments on the datasets provided by PINNacle (Hao et al., 2024). These experimental results enable us to leverage past experiences to solve new PDEs. Subsequently, we developed a new retrieval method named **Physics-Guided Knowledge Replay (PGKR)**, which extends the general Knowledge Replay concept by incorporating domain-specific knowledge. PGKR first encodes a PDE's mathematical and physical properties into a structured format. This allows the method to find retrieval PDEs with similar structures to the target PDE. By comparing the encoded representations, PGKR identifies the most relevant PDEs from the knowledge base, providing valuable insights into the appropriate PINNs architectures and hyperparameter settings.

To encode the mathematical and physical properties of PDEs into a structured format, we define a comprehensive set of labels $\mathcal{L} = \{l_1, l_2, \ldots, l_n\}$ that capture the key features of each PDE, including equation type (e.g., parabolic, elliptic, hyperbolic), spatial dimensions, linearity, time dependence, boundary and initial conditions, coefficient type, time scale, and geometric complexity. These labels are then encoded into feature vectors $\mathcal{F} = \{f_1, f_2, \ldots, f_n\}$ using a predefined encoding scheme. Based on findings from previous studies, we assign higher weights to certain critical features, particularly the PDE type, to enhance similarity determination. The encoding process can be formally defined as a function $\mathcal{E} : \mathcal{L} \to \mathcal{F}$, which maps each label to its corresponding weighted feature vector:

$$\mathcal{E}(l_i) = w_i f_i, \quad i = 1, 2, \ldots, n \qquad (6)$$

where $w_i$ is the weight assigned to the $i$-th feature.

The encoded feature vectors for all PDEs are concatenated to form a feature matrix $X \in \mathbb{R}^{n \times m}$, where $n$ is the number of PDEs and $m$ is the dimensionality of the feature space. Further details on the encoding scheme and weight assignment can be found in Appendix A.

To measure the similarity between PDEs, we employ a weighted cosine similarity, which quantifies the cosine of the angle between two weighted feature vectors. Given two PDEs represented by their feature vectors $f_i$ and $f_j$, the weighted cosine similarity $s_{ij}$ is computed as:

$$s_{ij} = \frac{(W f_i) \cdot (W f_j)}{\|W f_i\| \, \|W f_j\|}, \quad i, j = 1, 2, \ldots, n \qquad (7)$$

where $W$ is a diagonal matrix of weights. This weighting scheme allows us to emphasize the importance of certain features in determining similarity. The resulting similarity matrix $S \in \mathbb{R}^{n \times n}$ captures the pairwise similarities between all PDEs in the knowledge base. The top-$k$ most similar

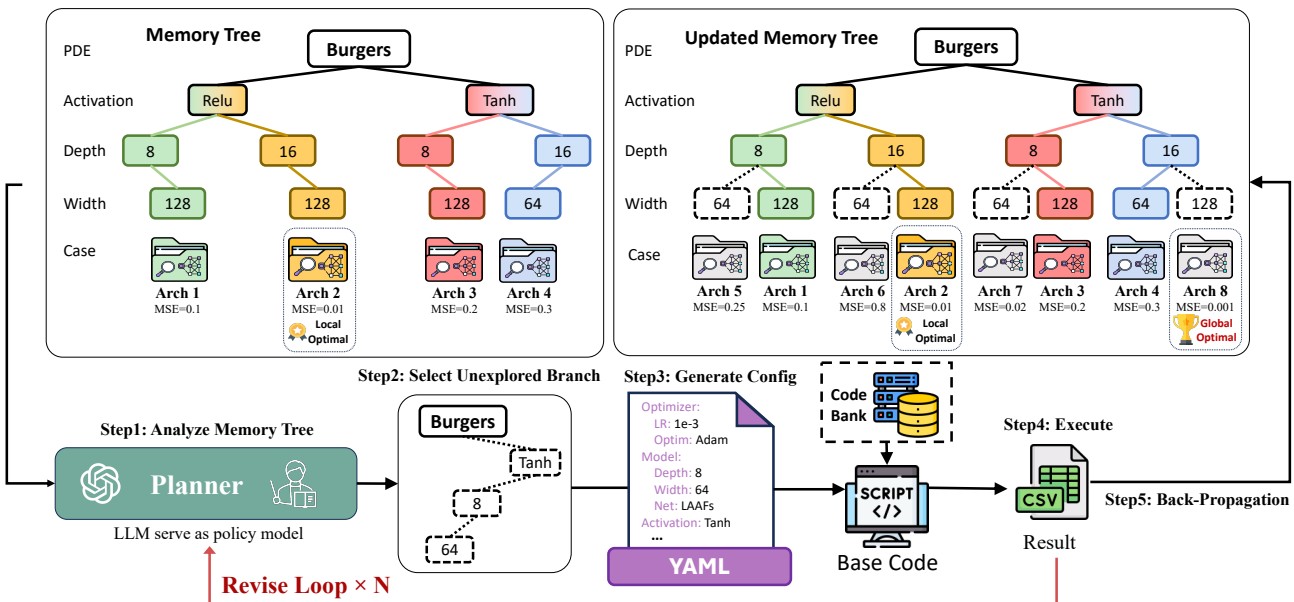

*Figure 2.* **Memory Tree Reasoning Strategy.** The root node represents the corresponding PDE, with subsequent levels corresponding to different hyperparameters. The planner selects unexplored branches and generates configurations, which are executed to obtain MSE scores. This process iterates to refine the tree and find the global optimal architecture (Arch 8 with the lowest MSE).

PDEs are retrieved by ranking the similarity scores, along with their associated best-performing PINNs configurations. These configurations serve as the starting points for the surrogate model search process.

### 3.4. Guided Exploration: Memory Tree Reasoning Strategy

Physics-Guided Knowledge Replay (PGKR) provides an effective sub-optimal hyperparameter configuration as an initial point. To further refine and optimize this configuration, inspired by Monte Carlo Tree Search (MCTS) (Browne et al., 2012), we introduce the **Memory Tree Reasoning Strategy (MTRS)** within *PINNsAgent*. The Memory Tree abstracts the hyperparameter optimization process, as illustrated in Figure 2, enabling the agent to utilize prior knowledge and feedback to guide the exploration of the Physics-Informed Neural Networks (PINNs) architecture space.

In the Memory Tree abstraction, the root node represents the PDE to be solved, and each subsequent level corresponds to a specific hyperparameter, such as optimizer or activation function. The child nodes within each level represent the possible values for the corresponding hyperparameter.

The Hyperparameter Optimization of PINNs can thus be formulated as a Monte Carlo Tree Search (MCTS) process. In this formulation,each node in the tree represents a state, denoted as $s_i$, which encapsulates the unique path to the root node $s_0$. The action $a_i$ at step $i$ involves selecting a specific hyperparameter from the subsequent layer. Consequently, each leaf node at the final layer represents a complete hyperparameter setting. The reward is simply designed as the negative Mean Squared Error (MSE) score of the selected setting. To leverage the LLM agent planner for guiding the expansion and exploration of the most promising nodes of the tree, we maintain a state-action value function $Q : \mathcal{S} \times \mathcal{A} \mapsto \mathbb{R}$, where $Q(s, a)$ estimates the expected future reward of taking action $a$ at state $s$.

#### 3.4.1. SELECTION

The first step is to select the most promising actions within the search space. To achieve this, we employ the well-known *Upper Confidence bounds applied to the Trees* (UCT) algorithm (Kocsis & Szepesvari, 2006):

$$a^* = \arg \max_{a \in \mathcal{A}(s)} \left[ Q(s, a) + \lambda \pi_{pl}(a|s) \sqrt{\frac{\ln N(s)}{N(s, a)}} \right], \quad (8)$$

where $N(s)$ is the number of times state $s$ has been visited, $N(s, a)$ is the number of times action $a$ is taken at node $s$, and $\lambda$ is a constant that balances exploration and exploitation. The planner, serving as the policy model $\pi_{pl}(a|s)$, uses the distribution of the LLM's output to determine the following action to take.

### 3.4.2. EXPANSION

This step expands the memory tree by adding new child nodes to the previous state. If the selected node is a terminal node, this step is skipped, and the process proceeds directly to the back-propagation step. We limit the range of selection to avoid generating unreasonable architecture.

### 3.4.3. SIMULATION

The planner iteratively selects new actions and expands the existing memory tree until terminal nodes are reached. During this process, the top-K and temperature values of the planner can balance the exploration and exploitation of the memory tree. For instance, the LLM's decisions are more diverse with higher temperatures.

### 3.4.4. BACK-PROPAGATION

After selecting an unexplored path, the planner generates a configuration, integrates it into the base code extracted from the code base, and obtains the execution results. At this stage, only the MSE score is needed to calculate the reward of $H^t$, defined as $\mathcal{R}(\mathcal{H}^t) = -L^{\text{test}}(u(\theta, \mathcal{H}^t), u_{\text{gt}})$. The back-propagation algorithm of MCTS is then executed to update the $Q(s, a)$ by aggregating the rewards from all future steps of the nodes along the path.

*Table 1.* Hyperparameter Search Spaces for PINNs Optimization Tasks

| Hyperparameter | Details |
|---|---|
| Net | FNNs, LAAFs, GAAFs |
| Activation | Elu, Selu, Sigmoid, SiLu, ReLU, Tanh, Swish, Sin, Gaussian |
| Width | 8 to 256 (Increment: 4) |
| Depth | 3 to 10 (Increment: 1) |
| Optimizer | SGD, Adam, MultiAdam, L-BFGS |
| Initializer | Glorot Normal/Uniform, He Normal/Uniform, Zeros |
| Learning Rate | $10^{-6}$ to $10^{-1}$ |
| Points (Dom/Bnd/Init) | 100 to 9600 (Increment: 500) |

## 4. Experiments

In this section, we discuss the experimental methodology used to evaluate the performance of our *PINNsAgent*.

Section 4.1 describes the experimental settings, including the dataset, hyperparameter search space, and baselines for comparison. In Section 4.2, we present the main results and

analyze the effectiveness of *PINNsAgent* in solving PDEs. Finally, Section 4.3 presents an ablation study to investigate the contributions of PGKR and the Memory Tree.

### 4.1. Experimental Settings

**Dataset.** We leverage the PINNacle benchmark dataset (Hao et al., 2024), a comprehensive collection of 20 representative PDEs spanning 1D, 2D, and 3D domains. These PDEs encompass diverse characteristics, including varying geometries, multi-scale phenomena, nonlinearity, and high dimensionality, providing a challenging testbed for evaluating PINNs architectures. Detailed descriptions are provided in Appendix B.

**Hyperparameter Search Space.** We extend the hyperparameter search space defined by (Wang et al., 2022; Wang & Zhong, 2024), with additional hyperparameters carefully curated from previous hyperparameter optimization (HPO) works (Klein & Hutter, 2019) to provide a more comprehensive exploration of the architectural landscape of PINNs. The configuration space, shown in Table 1, encompasses 4 architectural choices: network type, activation functions, width, and depth, along with 5 hyperparameters: optimizer, initializer, learning rate, loss weight coefficients, and domain/boundary/initial points.

**Task Description and Experimental Details.** We evaluate the performance of *PINNsAgent* on the task of Hyperparameter Optimization. In this task, *PINNsAgent* is required to optimize the hyperparameter configuration for a given PDE within 5 iterations. We implement *PINNsAgent* with GPT-4 model. To evaluate the ability of *PINNsAgent* to solve unseen PDEs, we did not provide the relevant database for the target PDE. During the implementation of PGKR, we selected $topk = 1$. For each PDE, we conducted ten repeated experiments and took the average of the lowest MSE to mitigate randomness, with a temperature of 0.7. We compare *PINNsAgent* with two baseline methods: (1) Random Search, a basic hyperparameter tuning method that selects configurations randomly, and (2) Bayesian Search, which uses Bayesian optimization to select configurations. We also provide PINNacle benchmark's best reported results as oracle/upper bound for reference.

### 4.2. Main Results

The comparative end-to-end performance of *PINNsAgent* and the baseline approaches on 14 different PDEs is presented in Table 2. The results demonstrate that *PINNsAgent* consistently outperforms the baselines, achieving the best performance on 12 out of 14 PDEs. Notably, *PINNsAgent* shows significant improvements over Random Search and Bayesian Search in complex PDEs such as NS-C, Heat-MS, and Heat-ND. For instance, on the NS-C equation,

*Table 2.* Comparative performance (MSE) of *PINNsAgent* and baseline approaches on 14 different PDEs for Task 1. Results are averaged over 10 runs to mitigate randomness. Values in parentheses represent standard deviations. The best performances are highlighted in **bold**. PINNacle benchmark's best reported results are shown in gray for reference.

| | PDEs | Random Search | Bayesian Search | PINNsAgent | PINNacle Benchmark |
|---|---|---|---|---|---|
| **1D** | **Burgers** | 6.63E-02 (±1.10E-01) | 8.70E-02 (±6.51E-03) | **6.51E-05** (±1.63E-05) | 7.90E-05 |
| | **Wave-C** | 1.50E-01 (±1.46E-01) | 1.78E-01 (±3.84E-02) | **3.33E-02** (±3.60E-02) | 3.01E-03 |
| | **KS** | **1.09E+00** (±3.58E-02) | 1.10E+00 (±2.55E-03) | **1.09E+00** (±3.20E-02) | 1.04E+00 |
| **2D** | **Burgers-C** | 2.48E-01 (±4.04E-03) | 2.42E-01 (±8.96E-03) | **2.04E-01** (±1.71E-02) | 1.09E-01 |
| | **Wave-CG** | **2.87E-02** (±4.98E-04) | 2.11E-02 (±1.12E-02) | 5.40E-02 (±7.89E-03) | 2.99E-02 |
| | **Heat-CG** | 3.96E-01 (±3.22E-01) | 1.17E-01 (±3.24E-02) | **1.80E-03** (±1.04E-03) | 8.53E-04 |
| | **NS-C** | 4.02E-03 (±5.93E-03) | 5.12E-03 (±1.33E-03) | **8.50E-06** (±6.80E-06) | 2.33E-05 |
| | **GS** | 4.28E-03 (±2.23E-05) | **4.03E-03** (±4.47E-04) | 4.32E-03 (±3.07E-05) | 4.32E-03 |
| | **Heat-MS** | 1.84E-02 (±1.18E-02) | 7.48E-03 (±3.81E-03) | **3.57E-05** (±2.3E-05) | 5.27E-05 |
| | **Heat-VC** | 3.57E-02 (±8.72E-03) | 3.93E-02 (±2.17E-03) | **5.52E-03** (±3.89E-03) | 1.76E-03 |
| | **Poisson-MA** | 5.87E+00 (±1.17E+00) | 5.82E+00 (±2.30E+00) | **3.16E+00** (±9.92E-01) | 1.83E+00 |
| **3D** | **Poisson-CG** | 3.82E-02 (±2.15E-02) | 2.55E-02 (±5.65E-03) | **1.59E-02** (±1.11E-02) | 9.51E-04 |
| **ND** | **Poisson-ND** | 1.30E-04 (±2.78E-04) | 4.72E-05 (±2.76E-06) | **2.09E-06** (±1.06E-05) | 2.09E-06 |
| | **Heat-ND** | 2.58E-02 (±9.87E-02) | 1.18E-04 (±8.92E-06) | **3.51E-07** (±7.92E-07) | 8.52E+00 |

*Table 3.* Ablation study: Comparative performance (MSE) of *PINNsAgent* variants using GPT-4 on 12 PDEs. Results are averaged over runs (mean ± std). Best performances are highlighted in **bold**.

| Method | Burgers | Wave-C | Burgers-C | Wave-CG | Heat-CG | NS-C | GS | Heat-MS | Heat-VC | Poisson-CG | Poisson-ND | Heat-ND |
|---|---|---|---|---|---|---|---|---|---|---|---|---|
| **PINNsAgent** | **6.51E-05** ±1.63E-05 | 3.33E-02 ±3.60E-02 | **2.04E-01** ±1.71E-02 | 5.40E-02 ±7.89E-03 | **1.80E-03** ±1.04E-03 | **8.50E-06** ±6.80E-06 | 4.32E-03 ±3.07E-05 | **3.57E-05** ±2.3E-05 | **5.52E-03** ±3.89E-03 | **1.59E-02** ±1.11E-02 | **2.09E-06** ±1.06E-05 | **3.51E-07** ±7.92E-07 |
| **w/o PGKR** | 7.41E-05 ±1.86E-05 | **2.87E-02** ±3.91E-02 | 2.17E-01 ±1.55E-02 | **3.19E-02** ±2.88E-03 | 1.38E-02 ±7.94E-03 | 1.53E-05 ±1.58E-05 | **4.31E-03** ±3.06E-05 | 3.85E-05 ±1.02E-04 | 6.19E-03 ±4.38E-03 | 2.27E-02 ±1.59E-02 | 2.02E-05 ±1.02E-01 | 4.92E-06 ±1.11E-05 |
| **w/o PGKR & MTRS** | 8.44E-05 ±5.97E-05 | 2.88E-02 ±3.32E-02 | 2.25E-01 ±1.87E-02 | 3.53E-02 ±1.29E-02 | 6.97E-02 ±2.07E-01 | 1.08E-05 ±8.65E-06 | 2.59E+08 ±7.77E+08 | 8.13E-05 ±1.43E-04 | 1.10E-02 ±1.36E-02 | 2.58E-02 ±1.68E-02 | 2.43E-05 ±2.47E-05 | 6.59E-07 ±1.01E-06 |

*PINNsAgent* achieves an MSE of 8.50E-06, which is several orders of magnitude better than Random Search (4.02E-03) and Bayesian Search (5.12E-03). These results highlight the effectiveness of *PINNsAgent* in optimizing PINNs architectures across a diverse range of PDEs.

These results confirm that the knowledge encoded and reused by *PINNsAgent* can be effectively transferred to different PDE classes and datasets, further highlighting the practical value of our approach for automated scientific machine learning.

### 4.3. Ablation Study

To gain a deeper understanding of the contributions of Physics-Guided Knowledge Replay (PGKR) and the Memory Tree Retrieval Strategy (MTRS) in *PINNsAgent*, we conducted an ablation study by removing these components individually and comparing the performance with the complete framework.

**Effectiveness of PGKR and MTRS.** Table 3 presents the performance of three variants of *PINNsAgent*: (1) the complete *PINNsAgent* framework, (2) *PINNsAgent* without PGKR (w/o PGKR), and (3) *PINNsAgent* without both PGKR and MTRS (w/o PGKR & MTRS).

The results demonstrate that both PGKR and MTRS contribute significantly to the performance of *PINNsAgent*. The complete *PINNsAgent* framework achieves the best performance on 9 out of 12 PDEs. Removing PGKR leads to performance degradation on most PDEs, with notable exceptions on Wave-C and Wave-CG. Further removing MTRS results in additional performance drops, most dramatically on the GS equation where the MSE increases from 4.31E-03 to 2.59E+08. These results validate the effectiveness of leveraging prior knowledge through PGKR and MTRS, demonstrating their crucial role in enhancing the performance and robustness of *PINNsAgent* across a diverse range of PDEs.

## 4.4. Computational Cost Analysis

We also conduct a computational cost analysis by reporting the average computation time per PDE for all methods. As shown in Table 4, the additional overhead introduced by LLM inference in *PINNsAgent* is only about 8.2% higher than Random Search and 4.1% higher than Bayesian Optimization.

*Table 4.* Average computation time (seconds) per PDE across all 14 benchmarks.

| Method | Average Computation Time (s) |
|---|---|
| Random Search | $3462.24 \pm 2631.55$ |
| Bayesian Search | $3598.47 \pm 2792.83$ |
| PINNsAgent | $3747.78 \pm 2965.62$ |

Our approach therefore remains efficient and practical for real-world scientific machine learning tasks.

## 5. Conclusion

In this work, we introduced *PINNsAgent*, a novel LLM-based surrogate framework that automates the development and optimization of PINNs for solving PDEs. By leveraging the knowledge and reasoning capabilities of large language models, *PINNsAgent* effectively bridges the gap between domain-specific knowledge and deep learning expertise, enabling non-experts to harness the power of PINNs without extensive manual tuning.

## Acknowledgments

This work was supported by the National Science and Technology Major Project (No. 2022ZD0117800).

## Impact Statement

We acknowledge several potential societal implications of *PINNsAgent*: (1) **Democratization vs. expertise dilution.** While *PINNsAgent* aims to make PINNs technology more accessible to non-experts, there is a concern that widespread automation might lead to insufficient understanding of the underlying physics and mathematical principles. We emphasize that *PINNsAgent* should serve as an educational tool that guides users toward better understanding rather than replacing the need for scientific rigor and domain expertise. (2) **API dependency and cost barriers.** The reliance on commercial LLM APIs creates potential barriers for researchers with limited funding or those in regions with restricted access to such services. We recommend exploring integration with open-source language models to ensure broader accessibility. (3) **Overconfidence in automated solutions.** There is a risk that automated optimization may create false confidence in PINNs solutions, particularly for complex physical systems. Users should maintain healthy skepticism about automatically generated configurations and always validate results against physical intuition and established numerical benchmarks before applying them to real-world problems with safety implications.

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

# Appendix

## A. PDE Feature Encoding and Similarity Analysis

### A.1. PDE Feature Encoding

To enable effective knowledge reuse in the PINNsAgent framework, we define a comprehensive set of labels $\mathcal{L} = l_1, l_2, \ldots, l_n$ that capture the essential mathematical and physical characteristics of each PDE problem. These labels include:

- PDE Type (Burgers, Poisson, Heat, NS, Wave, Chaotic)

- Equation Type (Parabolic, Elliptic, Hyperbolic, Mixed)

- Spatial Dimensions (1D, 2D, 3D, ND)

- Linearity (Linear, Nonlinear)

- Time Dependence (Time-Independent, Time-Dependent)

- Boundary Conditions (Dirichlet, Neumann, Mixed Robin, Periodic)

- Initial Conditions (Presence or Absence)

- Coefficient Type (Constant, Variable)

- Time Scale (Short-Time, Long-Time)

- Geometric Complexity (Simple, Complex)

We then encode these labels into binary feature vectors $\mathcal{F} = f_1, f_2, \ldots, f_n$, where $n$ is the number of PDEs, using a predefined encoding scheme, as shown in Table 5.

### A.2. PDE Labels

With the defined labels and encoding scheme, we assign a comprehensive set of labels to each PDE problem in our study. Table 6 and 7 present these labels for each PDE.

This encoding process can be formally defined as a function $\mathcal{E} : \mathcal{L} \to \mathcal{F}$, which maps each label to its corresponding binary feature vector. The encoded feature vectors for all PDEs are concatenated to form a feature matrix $X \in \mathbb{R}^{n \times m}$, where $n$ is the number of PDEs and $m$ is the dimensionality of the feature space. In our study, $n = 14$ and $m = 30$.

### A.3. PDE Similarity Analysis

Having obtained the PDE labels $\mathcal{L}$ and their corresponding feature vectors $\mathcal{F}$, we employ cosine similarity to quantify the similarity between PDEs , which measures the cosine of the angle between two feature vectors. The resulting similarity matrix $S \in \mathbb{R}^{n \times n}$ captures the pairwise similarities between all PDEs in the knowledge base.

## B. Datasets

We conduct our experiments using the datasets provided in PINNacle (Hao et al., 2024), which include 20 representative partial differential equations (PDEs) in one, two, and three dimensions. These equations exhibit a variety of complexities, including different geometries, multi-scale phenomena, nonlinearity, and high dimensionality.

For our study, we selected 14 PDEs from this collection. We excluded PInv and HInv because they are used for conducting inverse problems and Heat2d-LT and NS2d-LT because they present long-time PDEs that are poorly fitted by PINNs regardless of the hyperparameter configuration. Thus, their optimal hyperparameter configurations are not meaningful and can negatively impact the Physics-Guided Knowledge Replay (PGKR) approach during retrieval.

Below, we provide key properties of the PDEs used in this paper. For full details on the data generation process and the hyperparameters used to generate the PDE dataset, we refer the reader to (Hao et al., 2024).

Table 5. Feature encoding scheme for PDE labels

| Label | Description | Binary Encoding |
|---|---|---|
| PDE Type | Burgers | [1, 0, 0, 0, 0, 0] |
| | Poisson | [0, 1, 0, 0, 0, 0] |
| | Heat | [0, 0, 1, 0, 0, 0] |
| | Navier-Stokes | [0, 0, 0, 1, 0, 0] |
| | Wave | [0, 0, 0, 0, 1, 0] |
| | Chaotic | [0, 0, 0, 0, 0, 1] |
| Equation Type | Parabolic | [1, 0, 0, 0] |
| | Elliptic | [0, 1, 0, 0] |
| | Hyperbolic | [0, 0, 1, 0] |
| | Mixed | [0, 0, 0, 1] |
| Spatial Dimensions | 1D | [1, 0, 0, 0] |
| | 2D | [0, 1, 0, 0] |
| | 3D | [0, 0, 1, 0] |
| | ND | [0, 0, 0, 1] |
| Linearity | Linear | [1, 0] |
| | Nonlinear | [0, 1] |
| Time Dependence | Time-Independent | [1, 0] |
| | Time-Dependent | [0, 1] |
| Boundary Conditions | Dirichlet | [1, 0, 0, 0] |
| | Neumann | [0, 1, 0, 0] |
| | Mixed Robin | [0, 0, 1, 0] |
| | Periodic | [0, 0, 0, 1] |
| Initial Conditions | Initial Condition | [1, 0] |
| | No Initial Condition | [0, 1] |
| Coefficient Type | Constant Coefficient | [1, 0] |
| | Variable Coefficient | [0, 1] |
| Time Scale | Short-Time | [1, 0] |
| | Long-Time | [0, 1] |
| Geometric Complexity | Simple Geometry | [1, 0] |
| | Complex Geometry | [0, 1] |

## B.1. PDE Descriptions

### B.1.1. BURGERS EQUATION (1D)

The Burgers equation is a fundamental nonlinear partial differential equation (PDE) extensively used to model various fluid dynamics systems, including shock flows, wave propagation in combustion chambers, and vehicular traffic movement. The equation is given by:

$$\frac{\partial u}{\partial t} + u \frac{\partial u}{\partial x} - \frac{0.01}{\pi} \frac{\partial^2 u}{\partial x^2} = 0, \tag{9}$$

where $x \in [-1, 1]$ and $t \in [0, 1]$. Here, $\frac{0.01}{\pi}$ represents the diffusion coefficient of the fluid. The initial and boundary conditions are

$$u(x, 0) = -\sin \pi x,$$
$$u(-1, t) = u(1, t) = 0. \tag{10}$$

*Table 6.* PDE labels and their corresponding feature vectors

| PDE | Labels |
|---|---|
| Burgers1d | Burgers, parabolic, 1d, nonlinear, time-dependent, Dirichlet, initial-condition, constant-coefficient, short-time, simple-geometry |
| Burgers2d | Burgers, parabolic, 2d, nonlinear, time-dependent, periodic, initial-condition, constant-coefficient, short-time, simple-geometry |
| Poisson2d-C | Poisson, elliptic, 2d, linear, time-independent, dirichlet, no-initial-condition, constant-coefficient, short-time, complex-geometry |
| Poisson2d-CG | Poisson, elliptic, 2d, linear, time-independent, dirichlet, no-initial-condition, variable-coefficient, short-time, complex-geometry |
| Poisson3d-CG | Poisson, elliptic, 3d, linear, time-independent, neumann, no-initial-condition, variable-coefficient, short-time, complex-geometry |
| Poisson2d-MS | Poisson, elliptic, 2d, linear, time-independent, robin, no-initial-condition, variable-coefficient, short-time, complex-geometry |
| Heat2d-VC | Heat, parabolic, 2d, linear, time-dependent, dirichlet, initial-condition, variable-coefficient, short-time, simple-geometry |
| Heat2d-MS | Heat, parabolic, 2d, linear, time-dependent, dirichlet, initial-condition, constant-coefficient, short-time, simple-geometry |

### B.1.2. 2D COUPLED BURGERS EQUATION (BURGERS 2D)

The 2D Coupled Burgers equation, which extends the 1D Burgers equation to two dimensions, is expressed as:

$$\boldsymbol{u}_t + \boldsymbol{u} \cdot \nabla \boldsymbol{u} - \nu \Delta \boldsymbol{u} = 0, \tag{11}$$

where $(x, y) \in [0, 4]^2$ and $t \in [0, 1]$. Here, $\nu$ represents the diffusion coefficient of the fluid. The periodic boundary conditions are:

$$\boldsymbol{u}(0, y, t) = \boldsymbol{u}(L, y, t), \quad \boldsymbol{u}(x, 0, t) = \boldsymbol{u}(x, L, t) \tag{12}$$

The initial conditions are given by:

$$\boldsymbol{w}(x, y) = \sum_{i=-L}^{L} \sum_{j=-L}^{L} \boldsymbol{a}_{ij} \sin(2\pi(ix + jy)) + \boldsymbol{b}_{ij} \cos(2\pi(ix + jy))$$
$$\boldsymbol{u}(x, y, 0) = 2\boldsymbol{w}(x, y) + \boldsymbol{c} \tag{13}$$

where $\boldsymbol{a}_{ij}, \boldsymbol{b}_{ij}$, and $\boldsymbol{c}$ are normally distributed random variables with mean 0 and variance 1, i.e., $\boldsymbol{a}_{ij}, \boldsymbol{b}_{ij}, \boldsymbol{c} \sim N(0, 1)$.

### B.1.3. POISSON 2D CLASSIC (POISSON2D-C)

The Poisson 2D equation is widely used to describe various physical phenomena, such as electrostatics and fluid dynamics. It models the distribution of a scalar field $u(x, y)$ governed by Laplace's operator:

$$-\Delta u = 0. \tag{14}$$

Table 7. PDE labels and their corresponding feature vectors

| PDE | Labels |
| --- | --- |
| Heat2d-CG | Heat, parabolic, 2d, linear, time-dependent, robin, initial-condition, constant-coefficient, short-time, complex-geometry |
| NS2d-C | NS, mixed, 2d, nonlinear, time-independent, dirichlet, no-initial-condition, constant-coefficient, short-time, simple-geometry |
| NS2d-CG | NS, mixed, 2d, nonlinear, time-independent, dirichlet, no-initial-condition, constant-coefficient, short-time, complex-geometry |
| Wave1d-C | Wave, hyperbolic, 1d, linear, time-dependent, dirichlet, initial-condition, constant-coefficient, short-time, simple-geometry |
| Wave2d-CG | Wave, hyperbolic, 2d, linear, time-dependent, neumann, initial-condition, variable-coefficient, short-time, complex-geometry |
| Wave2d-MS | Wave, hyperbolic, 2d, linear, time-dependent, dirichlet, initial-condition, constant-coefficient, long-time, simple-geometry |
| GS | Chaotic, parabolic, 2d, nonlinear, time-dependent, no-boundary-condition, initial-condition, constant-coefficient, long-time, simple-geometry |
| KS | Chaotic, parabolic, 1d, nonlinear, time-dependent, periodic, initial-condition, constant-coefficient, short-time, simple-geometry |
| PNd | Poisson, elliptic, nd, linear, time-independent, dirichlet, no-initial-condition, constant-coefficient, short-time, simple-geometry |
| HNd | Heat, parabolic, nd, linear, time-dependent, mixed-boundary, initial-condition, constant-coefficient, short-time, simple-geometry |

We define the regions $R_1, R_2, R_3$, and $R_4$ as:

$$
\begin{aligned}
R_1 &= \{(x,y) : (x - 0.3)^2 + (y - 0.3)^2 \le 0.12\}, \\
R_2 &= \{(x,y) : (x + 0.3)^2 + (y - 0.3)^2 \le 0.12\}, \\
R_3 &= \{(x,y) : (x - 0.3)^2 + (y + 0.3)^2 \le 0.12\}, \\
R_4 &= \{(x,y) : (x + 0.3)^2 + (y + 0.3)^2 \le 0.12\}.
\end{aligned}
\tag{15}
$$

This equation operates within the spatial domain $\Omega = \Omega_{rec} \setminus \{R_i\}$, where $\Omega_{rec} = [-0.5, 0.5]^2$ denotes a rectangular region excluding the four circular areas $R_i$. The boundary condition is defined as:

$$
\begin{aligned}
u &= 0, x \in \partial R_i \\
u &= 1, x \in \partial \Omega_{\text{rec}} .
\end{aligned}
\tag{16}
$$

### B.1.4. POISSON-BOLTZMANN (HELMHOLTZ) 2D IRREGULAR GEOMETRY (POISSON2D-CG)

The Poisson-Boltzmann (Helmholtz) 2D equation extends the classic Poisson equation by including a term proportional to the scalar field $u$. The equation is given by:

$$-\Delta u + k^2 u = f(x, y). \tag{17}$$

The computational domain is defined as $[-1, 1]^2$ excluding several circular regions $\Omega_{\text{circle}} = \cup_{i=1}^4 R_i$, which are defined by:

$$
\begin{aligned}
R_1 &= \left\{ (x, y) : (x - 0.5)^2 + (y - 0.5)^2 \le 0.2^2 \right\}, \\
R_2 &= \left\{ (x, y) : (x - 0.4)^2 + (y + 0.4)^2 \le 0.4^2 \right\}, \\
R_3 &= \left\{ (x, y) : (x + 0.2)^2 + (y + 0.7)^2 \le 0.1^2 \right\}, \\
R_4 &= \left\{ (x, y) : (x + 0.6)^2 + (y - 0.5)^2 \le 0.3^2 \right\}.
\end{aligned}
\tag{18}
$$

The boundary conditions are:

$$
\begin{aligned}
u &= 0.2, \quad x \in \partial\Omega_{\text{rec}}, \\
u &= 1, \quad x \in \partial\Omega_{\text{circle}}.
\end{aligned}
\tag{19}
$$

### B.1.5. POISSON 3D COMPLEX GEOMETRY WITH TWO DOMAINS (POISSON3D-CG)

The Poisson 3D equation with two distinct regions is described by:

$$-\mu_i \Delta u + k_i^2 u = f(x, y, z), \quad i = 1, 2. \tag{20}$$

The computational regions are defined as follows:

$$
\begin{aligned}
\Omega_1 &= [0, 1] \times [0, 1] \times [0, 0.5] \setminus \bigcup_{i=1}^4 R_i, \\
\Omega_2 &= [0, 1] \times [0, 1] \times [0.5, 1] \setminus \bigcup_{i=1}^4 R_i.
\end{aligned}
\tag{21}
$$

The spherical regions $R_i$ are given by:

$$
\begin{aligned}
R_1 &= \left\{ (x, y, z) : (x - 0.4)^2 + (y - 0.3)^2 + (z - 0.6)^2 \le 0.2^2 \right\}, \\
R_2 &= \left\{ (x, y, z) : (x - 0.6)^2 + (y - 0.7)^2 + (z - 0.6)^2 \le 0.2^2 \right\}, \\
R_3 &= \left\{ (x, y, z) : (x - 0.2)^2 + (y - 0.8)^2 + (z - 0.7)^2 \le 0.1^2 \right\}, \\
R_4 &= \left\{ (x, y, z) : (x - 0.6)^2 + (y - 0.2)^2 + (z - 0.3)^2 \le 0.1^2 \right\}.
\end{aligned}
\tag{22}
$$

The boundary condition is:

$$\frac{\partial u}{\partial n} = 0, \quad x \in \partial\Omega. \tag{23}$$

### B.1.6. 2D POISSON EQUATION WITH MANY SUBDOMAINS (POISSON2D-MS)

The PDE and boundary condition are given by:

$$
\begin{aligned}
-\nabla \cdot (a(x)\nabla u) &= f(x, y), \quad x \in \Omega, \\
\frac{\partial u}{\partial n} + u &= 0, \quad x \in \partial\Omega.
\end{aligned}
\tag{24}
$$

Here, the domain is $(x, y) \in \Omega = [-10, 10]^2$. The entire domain is divided into many small squares, with $a(x)$ being a piecewise linear function within each square.

### B.1.7. 2D HEAT WITH VARYING COEFFICIENTS (HEAT2D-VC)

The heat equation models the distribution of temperature in a given region over time, describing how heat diffuses through a medium. The 2D heat equation with a varying source is given by:

$$\frac{\partial u}{\partial t} - \nabla(a(x)\nabla u) = f(x, t), \tag{25}$$

where the computational domain is $\Omega \times T = [0, 1]^2 \times [0, 5]$. The initial and boundary conditions are defined as:

$$\begin{aligned} u(x, y, 0) &= 0, x \in \Omega \\ u(x, y, t) &= 0, x \in \partial\Omega. \end{aligned} \tag{26}$$

### B.1.8. 2D HEAT EQUATION WITH MULTI-SCALE FEATURES (HEAT2D-MS)

The 2D heat equation with multi-scale features is described by:

$$\frac{\partial u}{\partial t} - \frac{1}{(500\pi)^2}u_{xx} - \frac{1}{\pi^2}u_{yy} = 0, \tag{27}$$

over the domain $\Omega \times T = [0, 1]^2 \times [0, 5]$. The initial and boundary conditions are given by:

$$\begin{aligned} u(x, y, 0) &= \sin(20\pi x)\sin(\pi y), \quad x \in \Omega, \\ u(x, y, t) &= 0, \quad x \in \partial\Omega. \end{aligned} \tag{28}$$

### B.1.9. 2D HEAT COMPLEX GEOMETRY (HEAT EXCHANGER, HEAT2D-CG)

The 2D heat equation for a complex geometry is given by:

$$\frac{\partial u}{\partial t} - \Delta u = 0. \tag{29}$$

The domain is defined as $\Omega \times T = ([-8, 8] \times [-12, 12] \setminus \cup_i R_i) \times [0, 3]$.

### B.1.10. 2D HEAT LONG TIME (HEAT2D-LT)

The long-time 2D heat equation is defined by:

$$\frac{\partial u}{\partial t} = 0.001\Delta u + 5\sin\left(ku^2\right)\left(1 + 2\sin\left(\frac{\pi t}{4}\right)\right)\sin\left(m_1\pi x\right)\sin\left(m_2\pi y\right), \tag{30}$$

where the computational domain is $\Omega \times T = [0, 1]^2 \times [0, 100]$. The initial and boundary conditions are given by:

$$\begin{aligned} u(x, y, 0) &= \sin(4\pi x)\sin(3\pi y), x \in \Omega \\ u(x, y, t) &= 0, \quad x \in \partial\Omega \end{aligned} \tag{31}$$

### B.1.11. 2D NAVIER-STOKES LID-DRIVEN FLOW (NS2D-C)

The 2D Navier-Stokes equations describe the motion of fluid substances and are fundamental in fluid dynamics. The governing equations for describing lid-driven cavity flow are defined as:

$$\begin{aligned} \boldsymbol{u} \cdot \nabla\boldsymbol{u} + \nabla p - \frac{1}{Re}\Delta\boldsymbol{u} &= 0, \quad x \in \Omega, \\ \nabla \cdot \boldsymbol{u} &= 0, \quad x \in \Omega, \end{aligned} \tag{32}$$

where $\Omega = [0, 1]^2$ is the domain. The boundary conditions are specified as follows:

$$\begin{aligned}
\boldsymbol{u}(\boldsymbol{x}) &= (4x(1-x), 0), & x &\in \Gamma_1, \\
\boldsymbol{u}(\boldsymbol{x}) &= (0, 0), & x &\in \Gamma_2, \\
p &= 0, & x &= (0, 0),
\end{aligned} \tag{33}$$

where the top boundary is denoted as $\Gamma_1$, and the left, right, and bottom boundaries are denoted as $\Gamma_2$.

### B.1.12. 2D BACK STEP FLOW (NS-CG)

For the 2D back step flow, the Navier-Stokes equations and boundary conditions are described as follows:

$$\begin{aligned}
\boldsymbol{u} \cdot \nabla \boldsymbol{u} + \nabla p - \frac{1}{Re} \Delta \boldsymbol{u} &= 0, \\
\nabla \cdot \boldsymbol{u} &= 0,
\end{aligned} \tag{34}$$

The domain is defined as $\Omega = [0, 4] \times [0, 2] \setminus ([0, 2] \times [1, 2] \cup R_i)$, which excludes the top-left quarter. The boundary conditions are:

$$\begin{aligned}
\boldsymbol{u}_{\text{in}} &= 4y(1 - y), \\
p &= 0, \quad \text{at the outlet}, \\
\boldsymbol{u} &= 0, \quad \text{no-slip condition},
\end{aligned} \tag{35}$$

### B.1.13. 1D BASIC WAVE EQUATION (WAVE1D-C)

The 1D wave equation is a second-order partial differential equation that describes the propagation of waves, such as sound or light waves, through a medium. The standard form of the wave equation in one dimension is given by:

$$u_{tt} - 4u_{xx} = 0 \tag{36}$$

The domain for this problem is $\Omega \times T = [0, 1] \times [0, 1]$. The boundary conditions are:

$$u(0, t) = u(1, t) = 0 \tag{37}$$

The initial conditions are:

$$u(x, 0) = \sin(\pi x) + \frac{1}{2}\sin(4\pi x) \, u_t(x, 0) \quad = 0 \tag{38}$$

### B.1.14. 2D WAVE EQUATION IN HETEROGENEOUS MEDIUM (WAVE2D-CG)

The governing PDE for the 2D wave equation in a heterogeneous medium is given by:

$$\left[ \nabla^2 - \frac{1}{c(x)} \frac{\partial^2}{\partial t^2} \right] u(x, t) = 0 \tag{39}$$

The domain is $\Omega = [-1, 1] \times [-1, 1]$, and the initial conditions are:

$$\begin{aligned}
u(x, 0) &= \exp\left( -\frac{|x - \mu|^2}{2\sigma^2} \right), \quad x \in \Omega \\
\frac{\partial u}{\partial t}(x, 0) &= 0, \quad x \in \Omega
\end{aligned} \tag{40}$$

The boundary condition is:

$$\frac{\partial u}{\partial n} = 0, \quad x \in \partial\Omega \tag{41}$$

### B.1.15. 2D DIFFUSION-REACTION GRAY-SCOTT MODEL (GS)

The Gray-Scott model describes the pattern formation in reaction-diffusion systems. The governing PDEs are:

$$
\begin{aligned}
u_t &= \varepsilon_1 \Delta u + b(1 - u) - uv^2 \\
v_t &= \varepsilon_2 \Delta v - dv + uv^2
\end{aligned} \tag{42}
$$

The domain is $\Omega \times T = [-1, 1]^2 \times [0, 200]$ . The initial conditions are:

$$
\begin{aligned}
u(x, y, 0) &= 1 - \exp\left(-80\left((x + 0.05)^2 + (y + 0.02)^2\right)\right) \\
v(x, y, 0) &= \exp\left(-80\left((x - 0.05)^2 + (y - 0.02)^2\right)\right)
\end{aligned} \tag{43}
$$

### B.1.16. 2D KURAMOTO-SIVASHINSKY EQUATION (KS)

The Kuramoto-Sivashinsky equation models the chaotic behavior in systems such as flame fronts and fluid surfaces. The governing PDE is:

$$u_t + \alpha u u_x + \beta u_{xx} + \gamma u_{xxxx} = 0 \tag{44}$$

The domain is $\Omega \times T = [0, 2\pi] \times [0, 1]$. The initial condition is:

$$u(x, 0) = \cos(x)(1 + \sin(x)) \tag{45}$$

### B.1.17. N-DIMENSIONAL POISSON EQUATION (PND)

The Poisson equation is a fundamental PDE in potential theory and electrostatics. The N-Dimensional Poisson Equation is given by:

$$-\Delta u = \frac{\pi^2}{4} \sum_{i=1}^{n} \sin\left(\frac{\pi}{2} x_i\right) \tag{46}$$

The domain is defined by $\Omega = [0, 1]^n$.

### B.1.18. N-DIMENSIONAL HEAT EQUATION (HND)

The heat equation describes the distribution of heat (or variation in temperature) in a given region over time. The N-Dimensional Heat Equation is:

$$
\begin{aligned}
\frac{\partial u}{\partial t} &= k\Delta u + f(x, t), \quad x \in \Omega \times [0, 1] \\
\boldsymbol{n} \cdot \nabla u &= g(x, t), \quad x \in \partial\Omega \times [0, 1] \\
u(x, 0) &= g(x, 0), \quad x \in \Omega
\end{aligned} \tag{47}
$$

The geometric domain $\Omega = \{x : |x|_2 \leq 1\}$ is a unit sphere in $d$-dimensional space.

## C. Pseudocodes

---

**Algorithm 1** Physics-Guided Knowledge Replay (PGKR) for a New PDE

---

1: **Input:** Knowledge base $\mathcal{K} = \{(P_i, l_i, C_i)\}_{i=1}^{n}$, where:
2:      - $P_i$ is the $i$-th PDE in the knowledge base,
3:      - $l_i$ is the label set of the $i$-th PDE,
4:      - $C_i$ is the best-performing PINNs configuration for the $i$-th PDE,
5:      - $n$ is the total number of PDEs in the knowledge base.
6: **Input:** New PDE $P_{\text{new}}$ with labels $\mathcal{L}_{\text{new}}$
7: **Output:** Top-$k$ most similar PDEs, their best-performing PINNs configurations, and the updated knowledge base
8: Initialize an empty set $\mathcal{S} = \{\}$ to store the similarities
9: Encode the labels of the new PDE: $f_{\text{new}} = \mathcal{E}(\mathcal{L}_{\text{new}})$
10: **for** $i = 1, 2, \ldots, n$ **do**
11:      Encode the labels of the $i$-th PDE in the knowledge base: $f_i = \mathcal{E}(l_i)$
12:      Compute the cosine similarity between the new PDE and the $i$-th PDE:
13:      $s_{\text{new},i} = \frac{f_{\text{new}} \cdot f_i}{\|f_{\text{new}}\| \|f_i\|}$
14:      Add the similarity to the set $\mathcal{S}$: $\mathcal{S} = \mathcal{S} \cup \{(P_i, s_{\text{new},i})\}$
15: **end for**
16: Sort the set $\mathcal{S}$ in descending order based on the similarity scores
17: Select the top-$k$ most similar PDEs: $\mathcal{T} = \{(P_j, s_{\text{new},j})\}_{j=1}^{k}$
18: Retrieve the best-performing PINNs configurations for the top-$k$ PDEs: $\mathcal{C} = \{C_j | (P_j, s_{\text{new},j}) \in \mathcal{T}\}$
19: Use the retrieved PINNs configurations $\mathcal{C}$ as starting points for solving the new PDE $P_{\text{new}}$
20: Fine-tune the PINNs configurations in $\mathcal{C}$ for the new PDE $P_{\text{new}}$
21: Select the best-performing PINNs configuration $C_{\text{new}}$ for the new PDE $P_{\text{new}}$
22: Add the new PDE, its feature vector, and its best-performing PINNs configuration to the knowledge base:
23: $\mathcal{K} = \mathcal{K} \cup \{(P_{\text{new}}, f_{\text{new}}, C_{\text{new}})\}$
24: **Return:** $\mathcal{T}$, $\mathcal{C}$, and the updated knowledge base $\mathcal{K}$

---

## D. Prompt

### D.1. Task Description

---

**Task Description**

[Task]

You are an expert in the field of neural architecture search (NAS) and physics-informed neural networks (PINNs). Your task is to suggest a hyperparameter configuration for solving the Burgers-1D equation, a Partial Differential Equation (PDE), using a PINNs model. The objective is to maximize the model's performance, measured by minimizing the Mean Squared Error (MSE) and running time.

The Burgers-1D equation is given by:
u_t + u u_x = nu u_xx

The domain is defined as:
(x, t) in Omega = [-1, 1] x [0, 1]

The initial and boundary conditions are:
u(x, 0) = -sin(pi x)
u(-1, t) = u(1, t) = 0

The parameter is:
nu = 0.01/pi

The search space includes various options for the network architecture, optimization algorithm, and other settings:

- "net" options: FNN (Fully-connected neural network), LAAF (Locally Adaptive Activation Functions), GAAF (Globally Adaptive Activation Functions)

---

- "optimizer" options: adam (Adam optimizer), multiadam (Multiscale Adam optimizer), lbfgs (Limited-memory BFGS algorithm)

The full search space is provided in the following JSON file:

[train.yaml]

```
{
  "name": "Transfer_Learning",
  "seed": 44,
  "log_every": 100,
  "plot_every": 2000,
  "repeat": 1,
  "iter": 20000,
  "pde_list": ['Burgers1D'],
  "activation": ["elu", "selu", "sigmoid", "silu", "relu", "tanh",
  "swish", "gaussian"],
  "net": ["fnn", "laaf", "gaaf"],
  "optimizer": ["adam", "multiadam", "lbfgs"],
  "loss_weight": ["none"],
  "width": [8, 16, 20, 24, 28, 32, 36, 40, 44, 48, 52, 56, 60, 64, 68, 72,
  76, 80, 84, 88, 92, 96, 100, 104, 108, 112, 116, 120, 124, 128, 132,
  136, 140, 144, 148, 152, 156, 160, 164, 168, 172, 176, 180, 184, 188,
  192, 196, 200, 204, 208, 212, 216, 220, 224, 228, 232, 236, 240, 244,
  248, 252, 256],
  "depth": [3, 4, 5, 6, 7, 8, 9, 10],
  "lr": [1e-6, 1e-5, 1e-4, 1e-3, 1e-2, 1e-1],
  "num_domain_points": [100, 600, 1100, 1600, 2100, 2600, 3100, 3600,
  4100, 4600, 5100, 5600, 6100, 6600, 7100, 7600, 8100, 8600, 9100,
  9600],
  "num_boundary_points": [100, 600, 1100, 1600, 2100, 2600, 3100, 3600,
  4100, 4600, 5100, 5600, 6100, 6600, 7100, 7600, 8100, 8600, 9100,
  9600],
  "num_initial_points": [100, 600, 1100, 1600, 2100, 2600, 3100, 3600,
  4100, 4600, 5100, 5600, 6100, 6600, 7100, 7600, 8100, 8600, 9100,
  9600],
  "initializer": ["Glorot normal", "Glorot uniform", "He normal",
  "He uniform", "zeros"]
}
```

## D.2. PGKR

### PGKR

Here are the most relevant hyperparameter configurations from previous experiments on PDEs that share similar characteristics with the PDE in the current [Task]. These configurations have shown good performance in the past, so they can serve as a valuable starting point for your hyperparameter search. However, keep in mind that the optimal configuration may still differ due to the unique properties of the current PDE. Use these configurations as a guide, but don't hesitate to explore other values within the provided search space.

```
[experiment_logs]
task  run time  mse  l2rel  mxe  crmse  activation net
optimizer  width  depth  lr domain_points  boundary_points
initial_points initializer  iter

Heat2D_VaryingCoef  1940.0  0.00165  0.205  0.255  sin  fnn   multiadam
```

```
100  5  0.001  8192  2048  2024  Glorot normal  NaN  0.00131
0.00174  20000
```

Please provide a single hyperparameter configuration that aims to improve performance by selecting one value for each key from the search space. Return the configuration as a properly formatted JSON object enclosed in triple backticks ("`).

## D.3. Optimization with the Guidance of Memory Tree

**Memory Tree**

Iteration 30:

```
Config: {'name': 'exp_1_Transfer_Learning', 'seed': 44, 'log_every': 100,
'plot_every': 2000, 'repeat': 1, 'iter': 200, 'pde_list': ['Burgers1D'],
'activation': 'swish', 'net': 'fnn', 'optimizer': 'adam', 'loss_weight':
'none', 'width': 120, 'depth': 8, 'lr': 0.001, 'num_domain_points': 9600,
'num_boundary_points': 9600, 'num_initial_points': 9600, 'initializer':
'He normal'}
```

MSE: 4.7700e-02 Run Time: 59.50s

Exploration Scores: activation: 1.1592 net: 0.9660 optimizer: 1.1592 width: 0.7728 depth: 0.7728 lr: 0.7728 num_domain_points: 0.2898 num_boundary_points: 0.2898 num_initial_points: 0.2898 initializer: 0.3864

Please provide an updated JSON configuration enclosed in triple backticks ("`) for the next iteration. Adjust the hyperparameters to improve both the Mean Squared Error (MSE) and Run Time.

Ensure to select values from the provided search space in [train.yaml].

Focus on tuning the following parameters: activation, optimizer, net. These parameters have the highest exploration scores, indicating they are the most promising for improving performance based on previous iterations.

Note that we have only a GPU with 24GB of memory. Return the configuration as a properly formatted JSON object enclosed in triple backticks.

