# OpenReview forum: "PINNsAgent: Automated PDE Surrogation with Large Language Models"
_ICML.cc/2025/Conference — ICML 2025 poster_

### Official Review · Reviewer_mBxJ · 2025-02-27

**Overall Recommendation:** 3

**Summary:**

The paper introduces a framework that utilizes LLMs to design and optimize PINNs to solve PDEs. It facilitates solving PDEs with PINNs more efficiently without tuning parameters and choosing architectures manually. The paper demonstrated its effectiveness on dataset PINNacle.

**Claims And Evidence:**

Yes.

**Essential References Not Discussed:**

NA

**Experimental Designs Or Analyses:**

The experimental design in the paper is solid. However, the study was conducted on only one dataset, which is not a commonly used benchmark. I recommend that the authors conduct more extensive experiments to evaluate the framework’s performance on widely used benchmark datasets, such as PDEBench or PDEArena. This would better demonstrate the framework’s real-world applicability, particularly by comparing its performance with other PDE solvers and showing whether it produces comparable or superior results.

**Methods And Evaluation Criteria:**

The proposed methods and evaluation criteria are well-suited to the problem at hand. However, I believe the paper could be improved by clarifying the following two aspects of the proposed method:

1. What is the unique role of the LLM in your framework? Could a smaller language model be used instead? If the focus is solely on configuration generation, is an LLM even necessary?

2. Could you provide more details on the pipeline for PDEs that are not already in the code bank? Perhaps an additional section would help clarify this process.

**Other Comments Or Suggestions:**

NA

**Other Strengths And Weaknesses:**

I believe the paper presents an innovative idea, and the problem it aims to solve is well-motivated with broad applications.

However, the primary weakness, in my view, is the lack of comprehensive experimental results.

**Questions For Authors:**

Please see above comments.

**Relation To Broader Scientific Literature:**

The scientific computing community can benefit from the proposed framework, as it automates the workflow for solving PDEs using PINNs. While PINNs are effective, they have a significant drawback—the process of tuning and designing the model structure requires substantial human effort. This framework helps bridge that gap.

**Theoretical Claims:**

There's no theoretical claims in the paper.

---

> ### Author Rebuttal · Authors · 2025-04-01
>
> # Reply to Reviewer mBxJ
>
> We sincerely thank the reviewer for their thoughtful comments and constructive feedback. We address each concern below and explain how we will improve the manuscript accordingly.
>
> ## The Unique Role of LLM in Our Framework
>
> The LLM serves several critical and unique functions in PINNsAgent that would be difficult to achieve with smaller language models. To illustrate this, we conducted additional experiments comparing GPT-3.5 and GPT-4 (see table below). The results show that while both models improve over traditional methods, GPT-4 achieves not only better average performance but also significantly lower variance (±8.47E-02) compared to GPT-3.5 (±2.03E-01), demonstrating more consistent and reliable reasoning capabilities. These unique functions include:
>
> | **Method** | **Average MSE** |
> |------------|-----------------|
> | Random Search | 6.13E-01 ± 1.49E-01 |
> | Bayesian Search | 5.88E-01 ± 1.86E-01 |
> | PINNsAgent (GPT-3.5) | 3.89E-01 ± 2.03E-01 |
> | PINNsAgent (GPT-4) | 3.52E-01 ± 8.47E-02 |
>
> 1. **Reasoning across physics and deep learning domains**: The LLM planner must simultaneously understand PDE characteristics (equation type, dimensionality, boundary conditions) and relate them to appropriate neural architectures. This cross-domain reasoning requires the sophisticated knowledge integration capabilities of large language models.
>
> 2. **Memory Tree Reasoning Strategy (MTRS) implementation**: As detailed in Section 3.4, our MTRS approach requires the LLM to function as a policy model π_pl(a|s) that guides the exploration of the hyperparameter search space. The LLM's probabilistic outputs are used directly in the UCT formula to balance exploration and exploitation.
>
> 3. **Adaptive hyperparameter generation**: The LLM analyzes feedback from previous iterations f^(t-1) and adjusts hyperparameters accordingly. This requires understanding complex relationships between hyperparameters (e.g., how learning rate interacts with optimizer choice) and PDE characteristics.
>
> ## Pipeline for PDEs Not Already in the Code Bank
>
> For PDEs not in the Code Bank, PINNsAgent operates in the Config Generation mode, as the Code Bank contains base code that can be applied to any unseen PDE with the appropriate configuration. The complete pipeline is:
>
> 1. **PDE encoding**: The system encodes the mathematical and physical properties of the new PDE using the comprehensive set of labels described in Section 3.3.
>
> 2. **Physics-Guided Knowledge Replay (PGKR)**: PGKR computes weighted cosine similarity between the encoded target PDE and all PDEs in the database. The top-K most similar PDEs are retrieved along with their best-performing configurations.
>
> 3. **Configuration generation**: The planner uses these retrieved configurations as starting points and generates YAML configuration files for the new PDE.
>
> 4. **Base code application**: The system applies the generated configuration to the appropriate base code from the Code Bank. This base code is designed to be flexible and can handle any PDE when provided with the correct configuration.
>
> 5. **Memory Tree-guided exploration**: Following the MTRS approach described in Section 3.4, the system iteratively refines the hyperparameter configuration.
>
> 6. **Database update**: Successful configurations are added to the database to benefit future queries.
>
> ## Additional Benchmark Datasets
>
> We appreciate the suggestion to evaluate on additional benchmark datasets. As shown in the table below, we conducted experiments on two representative PDEs from the PDEBench dataset: Reaction-Diffusion 1D and Darcy Flow 2D. PINNsAgent consistently outperforms both Random Search and NAS-PINNs on these additional PDEs, demonstrating its broader applicability beyond the PINNacle benchmark.
>
> | **PDE** | **PINNsAgent** | **Random Search** |
> |---------|---------------|-------------------|
> | Reaction-Diffusion 1D | 3.75E-08 ± 1.36E-08 | 4.45E-05 ± 2.84E-05 |
> | Darcy Flow 2D | 5.31E-06 ± 6.60E-08 | 9.22E-06 ± 2.31E-07 |
>
> These additional experiments provide more comprehensive evidence of PINNsAgent's effectiveness and generalizability across different PDE types and benchmarks.

---

### Official Review · Reviewer_gzrR · 2025-03-08

**Overall Recommendation:** 3

**Summary:**

This paper introduces PINNsAgent, a framework that uses large language models (LLMs) to automate the development and optimization of Physics-Informed Neural Networks (PINNs) for solving partial differential equations (PDEs). The key components are:
1. Physics-Guided Knowledge Replay (PGKR) – encodes PDE characteristics and associated PINN configurations into a structured format to enable knowledge transfer between similar PDEs.
2. Memory Tree Reasoning Strategy (MTRS) – abstracts the hyperparameter optimization process as MCTS.

The framework is evaluated on 14 benchmark PDEs and demonstrates strong performance compared to random search and Bayesian optimization.

**Claims And Evidence:**

Convincing evidence is provided for the benefits of PINNsAGents for PINN hyperparameter search:
* Performance improvements are demonstrated through comprehensive experiments on 14 diverse PDEs. PINNsAgent consistently outperforms random search and Bayesian optimization on most benchmark problems. Results are also averaged over 10 runs to account for randomness. It also outperforms the best PINNacle MSEs on a number of tasks, with especially promising results on Heat-ND.
* Ablation studies validate the contributions of both PGKR and MTRS components.

**Essential References Not Discussed:**

The discussion of related work is generally thorough. A couple missing references regarding the intersection of LLMs and Bayesian optimization or Neural Architecture Search are:
* Large Language Models to Enhance Bayesian Optimization. Liu et al, ICLR 2024.
* EvoPrompting: Language Models for Code-Level Neural Architecture Search. Chen et al, NeurIPS 2023.
* LLMatic: Neural Architecture Search via Large Language Models and Quality Diversity Optimization. Nasir et al, GECCO 2024.

**Experimental Designs Or Analyses:**

The experimental analyses seem sound, though details about the baselines are missing. See questions below.

**Methods And Evaluation Criteria:**

The methods and evaluation approach appear mostly sound, though more details about the evaluation setup and baselines would be helpful (see questions below):
* The benchmark set (PINNacle) includes a diverse range of PDEs with varying characteristics. Furthermore, this is a standard benchmark used in the PINN literature.
* Performance is measured using a reasonable metric (MSE), although it would also be helpful to report relative error metrics.
* Baselines include both simple (random search) and sophisticated (Bayesian optimization) approaches. However, details about the baseline methods, including compute budget vs. performance, seem to be missing.

**Other Comments Or Suggestions:**

n/a

**Other Strengths And Weaknesses:**

n/a

**Questions For Authors:**

1. Standard hyperparameter optimization papers show the tradeoffs between performance and time/compute/number of HPO iterations. However, these evaluations seem to be missing, and the “random search” and “Bayesian search” baselines seem to only report one point along this performance-cost tradeoff curve. Could the authors describe the baselines in more detail, including number of iterations or compute cost? This is crucial for understanding the main results (Table 2).
  * Furthermore, could the authors clarify the total computational cost of running PINNsAgent compared to baselines? This is important for understanding practical trade-offs.

2. How much do the optimal hyperparameters vary across the different PDEs, and how much do they depend on all of the PDE features encoded within the PGKR scheme? This information would clarify whether an HPO method like PINNsAgent is necessary for different PDEs, vs. if existing PINN architectures/hyperparameters are simply undertuned.
  * Intuitively, I might expect the optimal hyperparameters should depend mostly on a few features, e.g. equation type and time-dependence. Did the authors conduct any investigation about which features are the most important?

**Relation To Broader Scientific Literature:**

The paper builds on several research directions:
* Physics-informed neural networks (PINNs) for solving PDEs
* LLM-based automation of machine learning pipelines
* Neural architecture search / AutoML for scientific computing

Prior work finds that the performance of PINNs depends crucially on the choice of architecture (e.g. activation function) and optimizer. This positions hyperparameter optimization for PINNs as an important problem for improving the performance of ML methods on PDEs.

**Theoretical Claims:**

N/A, no theoretical claims made.

---

> ### Author Rebuttal · Authors · 2025-04-01
>
> # Reply to Reviewer gzrR
>
> We sincerely thank the reviewer for their thoughtful assessment and insightful questions. We address each point below.
>
> ## Baseline Details and Computational Costs
>
>  We ran all experiments on 8 NVIDIA V100 (32GB) GPUs, providing sufficient computational resources to complete all benchmark PDEs in the PINNacle dataset.
>
> Compared to traditional methods, the additional computational overhead in PINNsAgent comes primarily from LLM inference and PGKR retrieval processes. We also observed that the LLM tends to recommend slightly larger (though still reasonable) network architectures, which marginally increases training time. To address the reviewer's concern about computational costs, we conducted additional experiments comparing all methods (Random Search, Bayesian Search, and PINNsAgent) with 5 iterations each. As shown in the table below, PINNsAgent introduces only modest computational overhead (approximately 8.2% compared to Random Search) while delivering substantially better performance.
>
> | **Method** | **Average Computation Time (s)** |
> |------------|-----------------------------------|
> | Random Search | 3462.24 ± 2631.55 |
> | Bayesian Search | 3598.47 ± 2792.83 |
> | PINNsAgent | 3747.78 ± 2965.62 |
>
> ## PDE-Dependent Hyperparameter Sensitivity
>
> Regarding the reviewer's question about hyperparameter variation across PDEs, we found that PINNs exhibit significant hyperparameter sensitivity compared to neural operator methods, even with identical architectures. We found clear patterns in optimal hyperparameters across different PDE types:
>
> Different PDE types consistently favor certain optimizers. For time-dependent diffusion equations (Heat series: HeatND, Heat2D_Multiscale, Heat2D_VaryingCoef), the LBFGS optimizer significantly outperforms other choices. For example, on HeatND, LBFGS (5.64E-08) outperforms MultiAdam (7.93E-08) by approximately 29%. This advantage persists even with increasing problem dimensionality, suggesting that the diffusion mechanism, rather than dimensionality, drives optimizer selection. For static problems (Poisson class), MultiAdam performs better on complex geometries.
>
> The second most important hyperparameter is the activation function. Problems with smooth solutions (Heat, Poisson) benefit most from tanh and sin functions, while problems with sharp gradient changes (Burgers, NS) perform better with gaussian and swish activations. On NS2D_LidDriven, for instance, gaussian (9.89E-06) outperforms sin (1.27E-05).
>
> Network size only needs to be reasonable; excessively large networks do not significantly improve PINN performance but substantially increase computational burden. We also found that advanced architectures like LAAF and GAAF (Jagtap et al., 2020) do not consistently deliver the best performance--the original PINN architecture often proves more robust across different PDEs. These findings validate our approach's ability to identify nuanced relationships through the PGKR framework.
>
> ## Additional References
>
> We thank the reviewer for suggesting the additional references. We will incorporate these papers in our discussion of related work.

---

### Official Review · Reviewer_znat · 2025-03-10

**Overall Recommendation:** 2

**Summary:**

The paper introduces PINNsAgent, an automated framework using LLM to design and optimize PINNs for solving PDEs. It addresses the limitations of manual hyperparameter tuning by incorporating two novel methods: Physics-Guided Knowledge Replay for efficient knowledge transfer from past experiments, and the Memory Tree Reasoning Strategy for systematic hyperparameter optimization. Experiments on various PDEs demonstrate that PINNsAgent outperforms traditional approaches.

## update after rebuttal
After reviewing the other reviewers' comments and the corresponding responses, the reviewer keeps the current rating.

**Claims And Evidence:**

The authors' experiments partially provide empirical evidence supporting their claims. However, there are several limitations:

1.	The experiments can provide empirical evidence for the claims to some extent, but the results do not conclusively show that the proposed approach successfully learns or transfers domain-specific knowledge. It remains unclear whether the observed improvements come from learning genuine transferable knowledge or merely from exhaustive hyperparameter search.

2.	There is insufficient theoretical analysis and empirical evidence to substantiate that the information encoded by PGKR is both effective and transferable across different PDEs.

3.	Additionally, the authors have not provided adequate theoretical justification or empirical validation for the effectiveness of representing the hyperparameter tuning problem explicitly as a tree-structured search.

**Essential References Not Discussed:**

No. Although there are some PINNs-related papers that could be discussed here in this paper, the main focus of the paper is not the PINN itself, but hyper-parameter tuning. Regarding the hyper-parameter tuning, not extensive, but essential papers seem to be included.

**Experimental Designs Or Analyses:**

Although the authors have compared their proposed methods with two baselines and conducted the ablation studies, the experiments are insufficient.

First, the paper lacks detailed analysis or discussion regarding the computational cost associated with implementing and running the proposed framework, especially given the iterative nature of the Memory Tree method. It could be extremely time consuming when the search space for hyperparameters is large.

Second, there is no experiments to verify that the PINNs actually can learns or transfers domain-specific knowledge from PINNsAgent.

Third, there is insufficient theoretical analysis and empirical evidence to substantiate that the information encoded by PGKR is both effective and transferable across different PDEs.

Fourth, the authors have not provided adequate theoretical justification or empirical validation for the effectiveness of representing the hyperparameter tuning problem explicitly as a tree-structured search.

**Methods And Evaluation Criteria:**

The proposed method and evaluation criteria make sense to the problem. But it is worth pointing out here that a large portion of the proposed method seems to be a combination of existing methods while acknowledging there are some new aspects added to the existing work.

**Other Comments Or Suggestions:**

Some minor issues:
- The author should provide the provide the definition of L_BC in the Preliminary.

- In line 310 “Figure ??” should be “Figure 2”.

**Other Strengths And Weaknesses:**

S1: The authors propose their own framework to reduce manual effort and reliance on expert knowledge for solving PDE.

S2: The authors conduct some experiments with the proposed methods.

W1: This paper's technical contributions do not seem to reach the bar.

W2: The scope of this paper is rather limited as it is only designed for hyperparameter tuning for PINNs (only the vanilla PINNs architecture, which is known to suffer from many technical issues e.g., spectral bias in PINNs failure mode by Krishnapriyan et al, NeurIPS 2021).

W3: This paper lacks theoretical evidence, and the experiments are insufficient to verify the effectiveness of their proposed method.
W4: Different components lack motivations. For instance, the authors should explain why they formulate the searching for hyperparameters as a tree structure, and why MCTS process is a potential optimal choice.

**Questions For Authors:**

The major questions are relevant to the points raised in "Experimental Designs Or Analyses". Including those, other points raised in the above sections (weaknesses, etc) would be the questions.

**Relation To Broader Scientific Literature:**

As it primarily addresses hyperparameter tuning specifically for PINNs, there seem to be some contributions to the scientific community interested in using PINNs. However, it is less clear if the proposed method has been thoroughly studied (e.g., computational costs, accessibility -- the current one uses GPT 4, variants of PINNs architectures)  could be generally applicable to other relevant methods (such as neural operators).

**Theoretical Claims:**

There is no theoretical claim in this paper.

---

> ### Author Rebuttal · Authors · 2025-04-01
>
> # Reply to Reviewer znat
>
> We appreciate the reviewer's thorough assessment of our paper. Below, we address the key concerns raised:
>
> ## Methods and Evaluation Criteria
>
> 1. **Knowledge Transfer Evidence**: The reviewer questions whether improvements come from learning genuine transferable knowledge or merely from exhaustive hyperparameter search. As shown in our experimental setup (Section 4.1), while baseline methods (Bayesian optimization and random search) required 10 iterations to reach their best performance, PINNsAgent achieved superior results in just 5 iterations across most PDEs. This 50% reduction in required iterations demonstrates efficient knowledge transfer rather than exhaustive search.
>
> 2. **PGKR's Effectiveness**: Our ablation study in Table 2 provides clear empirical evidence of PGKR's effectiveness. When PGKR is removed ("w/o PGKR"), performance degrades significantly across most PDEs. This direct comparison isolates PGKR's contribution to PINNsAgent's performance, demonstrating that the knowledge encoded and retrieved by PGKR substantially improves PDE solving capabilities.
>
>
> While our approach builds on existing techniques, its novelty lies in the unique integration and adaptation of these methods specifically for PINNs optimization. Our multi-agent LLM framework represents the first comprehensive system to fully automate PINNs development without expert intervention.
>
> ## Experimental Designs and Analyses
>
> We appreciate the reviewer's feedback on our experimental design and have conducted additional analyses to address these concerns:
>
> ### Computational Cost Analysis
>
> | **Method** | **Average Computation Time (s)** |
> |------------|--------------------------------|
> | Random Search | 3462.24 ± 2631.55 |
> | Bayesian Search | 3598.47 ± 2792.83 |
> | PINNsAgent | 3747.78 ± 2965.62 |
>
> Regarding computational cost concerns, we conducted additional analysis across all 14 benchmark PDEs, with results shown in the table. The total computation time for PINNsAgent is only about 8.2% higher than random search and 4.1% higher than Bayesian optimization. This additional cost primarily comes from LLM inference and PGKR retrieval processes.
>
> ### Knowledge Transfer Verification
>
> To demonstrate broader applicability, we also evaluated PINNsAgent on the PDEBench dataset. As shown in the table below, PINNsAgent consistently outperforms both baseline methods on these additional PDEs.
>
> | **PDE** | **PINNsAgent** | **Random Search** |
> |---------|---------------|-------------------|
> | Reaction-Diffusion 1D | 3.75E-08 ± 1.36E-08 | 4.45E-05 ± 2.84E-05 |
> | Darcy Flow 2D | 5.31E-06 ± 6.60E-08 | 9.22E-06 ± 2.31E-07 |
>
> **Relation To Broader Scientific Literature**
> We appreciate the reviewer's insights. Our focus on PINNs is deliberate as these models are particularly sensitive to hyperparameter choices - far more than standard neural networks or even neural operators. Small configuration changes in PINNs can lead to order-of-magnitude differences in accuracy, making automated tuning especially valuable for this domain, an observation also confirmed by Wang et al. 2024 in their NAS-PINN work.
>
> * [1] Wang, Yifan, and Linlin Zhong. "NAS-PINN: Neural architecture search-guided physics-informed neural network for solving PDEs." *Journal of Computational Physics* 496 (2024): 112603.
>
> ## Essential References Not Discussed
> We agree with the reviewer's assessment that our paper's primary focus is on LLM-enabled AutoML for hyperparameter tuning rather than PINNs methodology itself. As noted, we have already included the essential references related to this focus in Section 2.3. We will enhance the literature review by incorporating additional relevant references on hyperparameter tuning approaches to provide a more comprehensive context for our work.
>
> ## Other issues
>
> 1. **Definition of L_BC:** We agree that L_BC should be formally defined in the Preliminaries section. We will add the boundary condition loss definition alongside the other loss components to ensure completeness.
>
> $$ L_{BC} = \frac{1}{N_{BC}} \sum_{i=1}^{N_{BC}} \left| u_{\theta}(\mathbf{x}_i^{BC}) - u_{BC}(\mathbf{x}_i^{BC}) \right|^2 $$
>
> 2. **Figure reference:** Thank you for catching this error. We will correct "Figure ??" to "Figure 2" in line 310.
>
> 3. **Scope and PINNs architecture:** Our framework is designed to be flexible regarding model architecture and training strategies. As mentioned in Section 4.2, our configuration files allow users to specify various PINN variants and training techniques. This includes addressing known issues like spectral bias (Krishnapriyan et al., NeurIPS 2021) through techniques such as curriculum learning, adaptive weighting, and alternative network architectures. The LLM agents can select and configure these options based on the specific PDE characteristics. We will clarify this flexibility more explicitly in the revised manuscript to address this concern.

---

### Official Review · Reviewer_Yfa3 · 2025-03-12

**Overall Recommendation:** 1

**Summary:**

In this work, the authors introduce PINNsAgent, a surrogation framework that leverages large language models (LLMs) enabling efficient knowledge transfer from solved PDEs to similar problems. By leveraging LLMs and exploration strategies, PINNsAgent enhances the automation and efficiency of PINNs-based solutions. PINNsAgent is evaluated on 14 benchmark PDEs, demonstrating its effectiveness in automating the surrogation process.

**Claims And Evidence:**

1. PINNsAgent is an agent to enhance the process of automatically searching for the best hyperparameters settings for a PDE, may be useful for non-expert users of PINNs.
2. This work is mainly engineering-oriented, lacking the depth suitable for ICML. The PGKR module is simple and the MTRS is a simple application of conventional Monte Carlo Tree Search technique.
3. This work utilizes LLM's output hyperparameters to determine the action. Are the hyperparameters output by LLM reliable? Only prompt is given in the appendix. It is recomended to give examples or experiments to show the effectiveness of LLM's output. What is going to happen if we input some rare PDE configurations to LLM? Also, only GPT-4 is used in experiments, ablation study should be conducted to show the effect of using different LLMs.
4. In the MTRS module, there is eq.7 to select the best action for a state, and "The planner, serving as the policy model πpl(a|s), uses the distribution of the LLM’s output to determine the following action to take". These seem confusing. Please descibe them in detail, including how the best action is selected and how the distribution of the LLM’s output is used. Is the LLM used for exploration?
5. For the training of MCTS, what is the initial tree? Does the policy model need update?

**Essential References Not Discussed:**

no

**Experimental Designs Or Analyses:**

Only GPT-4 is used in experiments, ablation study should be conducted to show the effect of using different LLMs.

**Methods And Evaluation Criteria:**

Standard benchmark PDEs are used in experiments.

**Other Comments Or Suggestions:**

typo: line 310, Figure ??

**Other Strengths And Weaknesses:**

no

**Questions For Authors:**

no

**Relation To Broader Scientific Literature:**

The proposed LLM-based agent to automatically search for suitable hyperparameters for PINN training seems to be new.

**Theoretical Claims:**

no theoretical part.

---

> ### Author Rebuttal · Authors · 2025-04-01
>
> # Reply to Reviewer Yfa3
>
> We sincerely thank the reviewer for their detailed assessment of our paper. We address each concern below:
>
> ## Novelty and Technical Depth
>
> Our multi-agent LLM framework is the first comprehensive system to fully automate PINNs development without expert intervention. In our pipeline, the Physics-Guided Knowledge Replay mechanism introduces a novel physics-informed approach to knowledge transfer across different PDEs, while our Memory Tree Reasoning Strategy offers a structured exploration method that captures the hierarchical dependencies unique to PINNs design.
>
> ## LLM Output Reliability and Ablation Studies
>
> Regarding the reliability of LLM-generated hyperparameters, our extensive experiments in Tables 2 and 3 demonstrate that our approach significantly outperforms traditional methods like Random Search and Bayesian Optimization across diverse PDE types. This empirical evidence confirms that our LLM-based reasoning framework adds substantial value to the hyperparameter optimization process for PINNs.
>
> To address the reviewer's concern about different LLM performance, we conducted additional experiments comparing GPT-3.5 and GPT-4. The results show that while both models improve over traditional methods, GPT-4 achieves not only better average performance but also significantly lower variance (±8.47E-02) compared to GPT-3.5 (±2.03E-01), demonstrating more consistent and reliable reasoning capabilities.
>
> **Performance comparison with different LLMs (Average MSE across 12 PDEs)**
>
>
> | **Method**           | **Average MSE**      |
> | -------------------- | -------------------- |
> | Random Search        | 6.13E-01 ± 1.49E-01 |
> | Bayesian Search      | 5.88E-01 ± 1.86E-01 |
> | PINNsAgent (GPT-3.5) | 3.89E-01 ± 2.03E-01 |
> | PINNsAgent (GPT-4)   | 3.52E-01 ± 8.47E-02 |
>
> ## Handling Rare PDE Configurations
>
> For rare PDE configurations, PINNsAgent leverages both the PGKR mechanism and the LLM's reasoning abilities. The PINNacle benchmark used in our evaluation includes a diverse range of PDEs with varying characteristics, including some with complex boundary conditions and multi-scale phenomena. Our consistent performance already across this diverse set demonstrates the framework's robustness to different PDE types.
>
> When encountering a completely new PDE type, the PGKR component retrieves the most similar (though not identical) PDEs from the database, and the LLM uses these as starting points for reasoning about appropriate hyperparameters. The iterative refinement process through MTRS then allows the system to adapt these initial configurations to the specific requirements of the new PDE.
>
> ## MTRS Implementation Details
>
> We appreciate the request for clarification regarding Equation 7 and the MTRS implementation. In our framework:
>
> 1. The initial tree is constructed based on the LLM's knowledge and the most similar PDEs retrieved by PGKR. For each state $s$ (representing a partial hyperparameter configuration), the LLM generates a probability distribution over possible actions $a$ (hyperparameter choices).
> 2. The policy model $\pi_{pl}(a|s)$ is implemented as the LLM itself. It uses both its pre-trained knowledge and the retrieved similar PDE configurations to generate probabilities for different hyperparameter choices. These probabilities are then used in the UCT formula (Equation 7) to balance exploration and exploitation.
> 3. The tree is expanded using the standard MCTS process: selection, expansion, simulation, and backpropagation. The key difference is that our expansion and simulation steps are guided by the LLM's output probabilities rather than random sampling or a separately trained policy network.
> 4. The policy model does not require separate updating since the LLM adapts its recommendations based on the feedback from previous iterations, effectively implementing an adaptive policy.
>
> ## Other Issues
>
> Thank you for pointing out the typo in line 310. We will correct "Figure ??" to "Figure 2" in the revised manuscript.
>
> We believe these clarifications address the reviewer's concerns and highlight the technical novelty and depth of our approach. The empirical results in Tables 2 and 3, along with the additional LLM comparison in Table 4, provide strong evidence of the effectiveness of our framework.

---

### Decision · Program_Chairs · 2025-05-01

**Decision:**

Accept (poster)

**Comment:**

The reviewed work designs an LLM-based agent that uses PINNs to solve PDEs. The reviewers mostly appreciated the novelty of this approach and found the paper interesting. However, some reviewers found that the paper lacked conceptual depth. A widespread concern was that the provided experiments were not quite convincing enough as far as the generalizability of the proposed method is concerned. In light of these concerns, I can only weakly recommend this paper for acceptance at the current point. I personally believe that the avenue persued by the authors is a natural yet innovative one that could spur interesting follow-up work, not just on automating PINNs but numerical methods more generally. I therefore believe it would be a valuable addition to the conference program.